# Sickle Cell Disease: Metabolomic Profiles of Vaso-Occlusive Crisis in Plasma and Erythrocytes

**DOI:** 10.3390/jcm9041092

**Published:** 2020-04-11

**Authors:** Klétigui Casimir Dembélé, Charlotte Veyrat-Durebex, Aldiouma Guindo, Stéphanie Chupin, Lydie Tessier, Yaya Goïta, Mohamed Ag Baraïka, Moussa Diallo, Boubacari Ali Touré, Chadi Homedan, Delphine Mirebeau-Prunier, Gilles Simard, Dapa Diallo, Bakary Mamadou Cissé, Pascal Reynier, Juan Manuel Chao de la Barca

**Affiliations:** 1Faculté de Pharmacie, Université des Sciences, des Techniques et des Technologies de Bamako, Bamako BP 1805, Mali; kletigui@outlook.fr (K.C.D.); aldguindo@icermali.org (A.G.); ahamedyaya@hotmail.fr (Y.G.); moagba08@yahoo.fr (M.A.B.); bakarymcisse@gmail.com (B.M.C.); 2Centre de Recherche et de Lutte contre la Drépanocytose, Bamako BP 1805, Mali; mdiallo888@yahoo.fr (M.D.); dianke74@gmail.com (B.A.T.); dadiallo@icermali.org (D.D.); 3Département de Biochimie et Génétique, Centre Hospitalier Universitaire, 49933 Angers, France; Stephanie.Chupin@chu-angers.fr (S.C.); LyTessier@chu-angers.fr (L.T.); ChHomedan@chu-angers.fr (C.H.); DePrunier@chu-angers.fr (D.M.-P.); GiSimard@chu-angers.fr (G.S.); JMChaoDeLaBarca@chu-angers.fr (J.M.C.d.l.B.); 4Unité Mixte de Recherche MITOVASC, Centre National de la Recherche Scientifique (CNRS) 6015, Institut National de la Santé et de la Recherche Médicale (INSERM) U1083, Université d’Angers, 49933 Angers, France; 5Unité Mixte de Recherche, Institut National de la Santé et de la Recherche Médicale (INSERM) U1253, iBRAIN, Université de Tours, 37000 Tours, France; charlotte.veyrat@live.fr; 6Service de Biochimie et Biologie Moléculaire, Centre Hospitalier Universitaire, 37000 Tours, France

**Keywords:** lipidomics, metabolomics, sickle cell disease, vaso-occlusive crisis

## Abstract

The metabolomic profile of vaso-occlusive crisis, compared to the basal state of sickle cell disease, has never been reported to our knowledge. Using a standardized targeted metabolomic approach, performed on plasma and erythrocyte fractions, we compared these two states of the disease in the same group of 40 patients. Among the 188 metabolites analyzed, 153 were accurately measured in plasma and 143 in red blood cells. Supervised paired partial least squares discriminant analysis (pPLS-DA) showed good predictive performance for test sets with median area under the receiver operating characteristic (AUROC) curves of 99% and mean *p*-values of 0.0005 and 0.0002 in plasma and erythrocytes, respectively. A total of 63 metabolites allowed discrimination between the two groups in the plasma, whereas 61 allowed discrimination in the erythrocytes. Overall, this signature points to altered arginine and nitric oxide metabolism, pain pathophysiology, hypoxia and energetic crisis, and membrane remodeling of red blood cells. It also revealed the alteration of metabolite concentrations that had not been previously associated with sickle cell disease. Our results demonstrate that the vaso-occlusive crisis has a specific metabolomic signature, distinct from that observed at steady state, which may be potentially helpful for finding predictive biomarkers for this acute life-threatening episode.

## 1. Introduction

The recurrent episodes of vaso-occlusive crises in sickle cell patients result from hemoglobin S polymerization, which increases the rigidity of the red blood cells. These rigid sickle erythrocytes have altered rheological and adhesive properties and become caught up in microcirculation, leading to microvascular obstructions in organs. Although the clinical manifestations of this acute episode are variable from one patient to another, the vaso-occlusive crisis is marked by intense pain and severe organ damage. The painful episode is life-threatening and the recurrence of crises causes disabling chronic complications, explaining the reduced life expectancy of these patients. The polymerization of hemoglobin S is induced by hypoxia, through anemia, fever, cold, dehydration, and any form of stress, including adverse socio-economic and environmental conditions [1]. The management of the vaso-occlusive crisis is mainly achieved by blood transfusion, hyperhydration, reoxygenation, and appropriate analgesia.

Biologically, the vaso-occlusive crisis is accompanied by hypercoagulability, as well as by deep metabolic, inflammatory, and oxidative stresses. More than 100 biomarkers have been reported in sickle cell disease, such as the disruption of the arginine and nitric oxide (NO) metabolism at the origin of sickle cell vasculopathy [2,3]. Despite these profound biological changes accompanying the crisis, relatively few metabolomics studies, which were recently reviewed [4], have been conducted in sickle cell patients, and to our knowledge, none of them have specifically reported the biological fingerprint of the crisis, in comparison to the steady state of the disease.

The most documented study, performed by Darghouth et al. [5], compared the red blood cell metabolomes of 28 adult patients with the HbSS (Hemoglobin SS) genotype at steady state and 24 healthy adults (HbAA). Among the 89 metabolites identified, 31 exhibited significantly modified concentrations, revealing the involvement of important metabolic pathways and metabolites including glycolysis, pentose phosphate, glutathione, ascorbate, amino acids, polyamines, carnitine, and creatine. Using metabolomic profiling of the whole blood and plasma in a mouse model of the disease (*n* = 6 per group), Xia et al., progressively deciphered a central pathogenic mechanism, also found in humans, promoting the polymerization of the hemoglobin S through sphingosine-1-P [6,7]. Their initial study revealed 251 significantly altered metabolites, involving glycolysis, pentose phosphate, amino acids, nucleotides, xenobiotics, lipids, fatty acids, and carbohydrate pathways. Lastly, a metabolomic approach was applied in the plasma of patients (*n* = 23) with sickle cell disease according to their level of albuminuria [8]. The concentration of six metabolites was found to be altered according to albuminuria: betaine, proline, dimethylarginine, glutamate, leucine, and lysine.

In this study, we used a standardized targeted metabolomics approach in order to explore the plasma and erythrocyte profiles of patients during their vaso-occlusive crisis, in comparison with the profiles obtained in the same patients after they return to the steady state a few months later.

## 2. Experimental Section

### 2.1. Ethics Statement

The study was conducted in accordance with the Declaration of Helsinki (1983) and was approved by the institutional ethical committee of the “Centre de Recherche et de Lutte Contre la Drépanocytose (CRLD)” of Bamako, Mali (N°18/002/MSHP-CRLD-CEI; 7 February 2018). Each participant gave his/her signed informed consent after receiving all the information necessary to understand the research protocol.

### 2.2. Study Participants

Figure 1 shows the global flow chart of the study design. The study included 40 individuals aged from 6 to 51 years old (mean = 18.64 years, 19 males and 21 females). Follow-up for all participants included was carried out at the “Centre de Recherche et de Lutte Contre la Drépanocytose” of Bamako, Mali. Participants were recruited between February and July 2018. The inclusion criteria were a confirmed diagnosis of sickle cell disease by HPLC. Individuals who had received a transfusion were excluded from the study. All patients were receiving analgesic treatments. At inclusion, during the crisis period, the average body temperature of patients was 37.6 °C (range 35.9–39.2 °C), their hemoglobin blood concentration was 9.01 g/dL on average (range 8.1–9.8), and their oxygen saturation (SpO_2_) was 95.25% on average (range: 83–99%). The patients were their own controls, as a new blood sample was taken after returning to a steady state a few (2–3) months later in the same specialized care center. The steady state for these patients corresponded to an absence of acute pain and fever, as well as any other signs or acute complication suggestive of a vaso-occlusive crisis for at least 2 months. The fasting blood samples were collected in the morning in heparinized tubes. They were immediately centrifuged at 3000 rpm for 10 min at +4 °C and 50 microliter aliquots of plasma, and red blood cells were stored at −80 °C before being transported in dry ice to the laboratory of biochemistry and molecular biology at the University Hospital of Angers, France, for the metabolomic analysis. After removing one sample with hemolysis, the plasma study involved 39 patients, whereas the erythrocyte study involved all the 40 patients.

### 2.3. Metabolomics Analyses

Red blood cell samples (50 μL) were mixed with a cold solution of phosphate-buffered saline (PBS; 15 μL) and methanol (85 µL), and were transferred to pre-cooled 2.0 mL homogenization Precellys tubes prefilled with 1.4 mm diameter ceramic beads. Homogenization was performed with two grinding cycles, each at 6500 rpm for 20 s, spaced by 20 s, followed by a grinding cycle at 6000 rpm for 30 s, using a Precellys homogenizer (Bertin Technologies, Montigny-le-Bretonneux, France) kept in a room at +4 °C. The supernatant (cell extract) was recovered after centrifuging the homogenate (10,000× *g*, 5 min at +4 °C) and kept at −80 °C until metabolomic analysis.

Targeted quantitative metabolomics analyses were performed using Biocrates AbsoluteIDQ p180 kit (Biocrates Life Sciences AG, Innsbruck, Austria) and a QTRAP 5500 mass spectrometer (SCIEX, Villebon-sur-Yvette, France). This kit allows quantification of up to 188 different endogenous molecules distributed as follows: free carnitine (C0), 39 acyl-carnitines (C), the sum of hexoses (H1), 21 amino acids, 21 biogenic amines, and 105 lipids. The lipids are distributed in the kit in four different classes: 14 lysophosphatidylcholines (LPC), 38 diacyl phosphatidylcholines (PC aa), 38 acyl-alkyl-phosphatidylcholines (PC ae), and 15 sphingomyelins (SM). Flow injection analysis coupled with tandem mass spectrometry (FIA-MS/MS) was used for analyzing carnitine, acyl-carnitines, lipids, and hexoses. Liquid chromatography (LC) was used to separate amino acids and biogenic amines before quantitation with mass spectrometry.

All reagents used in this analysis were of LC-MS grade and were purchased from VWR (Fontenay-sous-Bois, France) and Merck (Molsheim, France). The samples were prepared and analyzed following the Biocrates Kit User Manual. Briefly, each plasma or red blood cell extract sample was thoroughly vortexed after thawing, then centrifuged at +4 °C for 5 min at 5000× *g*. Next, 10 microliters of each sample were added to the filter on the upper wells of the 96-well plate. Metabolites were extracted and derivatized for quantitation of amino acids and biogenic amines. The extracts were finally diluted with MS running solvent before FIA- and LC-MS/MS analysis. Three quality controls (QCs) composed of human plasma samples at three concentration levels—low (QC1), medium (QC2), and high (QC3)—were used to evaluate the performance of the analytical assay. A seven-point serial dilution of calibrators was added to the 96-well plate of the kit to generate calibration curves for quantification of amino acids and biogenic amines.

### 2.4. Statistical Analyses

Before statistical analyses, the raw metabolomics data were examined to exclude metabolites with concentration values >20% below the lower limit of quantitation (LLOQ) or >20% above the upper limit of quantitation (ULOQ). To prevent the removal of discriminant features for metabolites having >20% of its concentration out of range, a chi-squared test was performed to determine the relation between the number of values within and without the quantitation range and crisis status.

Supervised principal component analysis (PCA) was used to detect similar groups of samples and outliers, that is, samples displaying an atypical metabolite profile. Partial least squares-discriminant analysis (PLS-DA) was performed on unit-variance scaled data to discriminate between in-crisis (IC) and out-of-crisis (OC) samples on the basis of their metabolic profiles. Data were randomly divided between the training-validation set comprising 30 in-crisis/30 out-of-crisis patients (3/4 of all samples) and the test set containing the remaining one-quarter of all samples (10 in-crisis/10 out-of-crisis samples for red cells and 9 in-crisis/9 out-of-crisis samples for plasma). The training-validation sets were used for constructing PLS-DA models and the test sets were used to validate the supervised model. Using cross-validation, there are $C_{20}^{30}=30\prime 045\prime 015$ combinations of 20 samples chosen out of 30, representing more than 30 million training with their corresponding validation sets. To make the cross-validation process computationally feasible, we stored all these combinations in a **M** = 20 × 30,045,015 matrix, then 23,345 combinations (i.e., models) were chosen by sampling **M** every 1287 columns. For each of the 23,345 models, the area under the receiving operator characteristic (AUROC) along with the *p*-value were calculated on the validation set. AUROCs are an almost universal indicator of the performance of a classifier, and AUROC values larger than 0.8 characterize models with good predictive capabilities, as long as the AUROCs are obtained in a set not used for constructing the supervised model. Each *p*-value associated with an AUROC indicates the probability of the equality of the supervised model and a random model (i.e., AUROC = 0.5).

Best models (i.e., AUROC ≥ 0.95) were chosen as the final models and their predictive capabilities were further tested in the test set. The global performance of PLS-DA models on our data was considered satisfactory only if the median AUROC and *p*-values were at least 0.8 and at most 0.05, respectively, when the best models of the training set were applied to the test set. In this case, discriminant variable selection was based on the variable importance for the projection (VIP) and the loading parameters. The VIP summarizes the importance of each variable for the PLS-DA model, whereas the loading indicates the relationship between the **y** vector containing the class information (i.e., in-crisis or out-of-crisis) and variables in the **X** matrix (i.e., metabolites). Variables with a VIP value greater than unity are important for group discrimination. Multivariate analysis was performed using the mixOmics R package (http://www.Rproject.org).

## 3. Results

After the validation of QCs and elimination of metabolites with concentration values >20% below the lower limit of quantitation or >20% above the upper limit of quantitation, 153 (81%) and 143 (76%) metabolites were considered accurately measured in the plasma and red blood cells, respectively, and were subsequently used for statistical analyses. These raw data (11,934 metabolite concentrations in the plasma and 11,440 in red blood cells) are available in Appendix A found in the Appendix A.

### 3.1. Metabolite Profile of the Vaso-Occlusive Crisis in the Plasma

The scatter plot of the unsupervised paired PCA did not show any spontaneous grouping or any strong outliers in the first principal plan (PC1 vs. PC2; Figure 2A). Paired PLS-DA showed good overall predictive performance of the supervised models with median AUROC and *p*-values in the validation set of 0.86 and 0.0065, respectively (Appendix A). More than 70% (16,516) of all models had AUROC ≥ 0.8. By selecting best final models (BMs) as those with the higher AUROC (i.e., AUROC ≥ 0.95) on the validation sets, 4277 (≈18%) were retained and their performance predictive capabilities (i.e., AUROC) were assessed on the test set. Very good performance capabilities were found when applying the BMs to the test set with median AUROC and *p*-values of 0.99 and 0.0005, respectively (Appendix A). Median coordinates from 4277 BMs were calculated for each sample used in the training-validation set, and the resulting scatter plot of the two latent variables showed good discrimination between in-crisis and out-of-crisis groups (Figure 2B).

Median VIP and loading were combined in a volcano plot (Figure 3). The best discriminant metabolites (i.e., VIP ≥ 1 and high absolute loading values) contributing to the model included a subset of 63 (41%) of the accurately measured metabolites in the plasma, comprising free carnitine, 2 acyl-carnitines, 13 amino acids, 6 biogenic amines, 1 sphingomyelin, 5 lysophosphatidylcholines, 12 diacyl phosphatidylcholines, 22 alkyl-acyl-phosphatidylcholines, and the sum of hexoses.

### 3.2. Metabolite Profile of the Vaso-Occlusive Crisis in the Red Blood Cells

The scatter plot of the paired PCA for red cell samples did not show any strong outliers but, in contrast with plasma samples, it showed a spontaneous separation of the two groups (PC1 vs. PC2; Figure 4A). Paired PLS-DA exhibited good predictive performance of the supervised model with median AUROC and *p*-values in the validation set of 0.91 and 0.0019, respectively (Appendix A). More than 83% (19,544) of models built with red cell samples had AUROC ≥ 0.8. By selecting best final models as those with the higher AUROC (i.e., AUROC ≥ 0.95) when applied to the validation sets, 7955 (34%) were retained, and their performance predictive capabilities (i.e., AUROC) were assessed on the test set. Very good performance capabilities were found when applying the BMs to the test set with median AUROC and *p*-values of 0.99 and 0.0002, respectively (Appendix A). Median coordinates from the 7955 BMs were calculated for each sample used in the training-validation set, and the resulting scatter plot of the two latent variables showed good discrimination between the in-crisis and out-of-crisis groups (Figure 4B).

Median VIP and loading were combined in a volcano plot (Figure 5). The best discriminant metabolites (i.e., VIP ≥ 1 and high absolute loading values) contributing to the model included a subset of 61 (43%) of the accurately measured metabolites in red blood cells. Lipid remodeling dominated the metabolomic signature with up to 42 phosphatidylcholine (PC) species (17 diacyl-PC or PC aa and 25 alkyl-acyl PC or PC ae) that increased during the crisis, along with 4 sphingomyelins and 5 lysophosphatidylcholines, all of them with an acyl chain of more than 22 carbons, and malonyl-carnitine. On the contrary, glutamine, aspartate, trans-4-hydroxyproline, serotonin, tetradecadienyl-carnitine, and four LPCs, all of them with an acyl chain length of less than 22 carbons, were found to be decreased during the crisis.

## 4. Discussion

Our results demonstrate that the vaso-occlusive crisis is associated with a specific metabolomics signature, whereby (i) the plasma mainly shows alteration of polar metabolites while several phospholipid species decrease, and (ii) erythrocytes show extensive phospholipids remodeling, with key polar metabolites also affected.

### 4.1. Plasmatic Metabolic Signature

The plasmatic signature is characterized by a decreased concentration in symmetric and asymmetric dimethylarginine (asymmetric dimethylarginine (ADMA), symmetric dimethylarginine (SDMA), and total DMA) as well as 13 amino acids (arginine, ornithine, citrulline, proline, methionine, asparagine, glutamine, alanine, threonine, lysine, glycine, serine, histidine), and an increase in phenylalanine. It is worth noting that DMA and most of these amino acids are linked to arginine and nitric oxide (NO) metabolism. In effect, the steady state of the disease is known to be associated with a decrease in arginine concentration because of arginine consumption to synthesize ornithine, polyamines, proline, and glutamate (via arginase) rather than NO (via NO synthase) [9,10]. During hemolysis, arginase is liberated from red blood cells and competes with NO synthases for arginine consumption and, furthermore, free hemoglobin fixes NO. This effect is exaggerated by ADMA, which originates from protein and tissue hydrolysis and acts as a strong competitive inhibitor of NO synthases. In fact, increased ADMA concentration in blood has been positively related to fatal outcomes in sickle cell disease [8,11]. Taken as a whole, the alteration of arginine and NO metabolism is an important feature of sickle cell disease that likely explains the observed changes in amino acid content.

Interestingly, arginine concentration, which is already affected in the steady state, is found to be decreased further by the vaso-occlusive crisis. By contrast, SDMA and ADMA, which are significantly higher in the steady state, decrease significantly during the crisis. This finding is in agreement with previous measurements of arginine and DMA in the plasma of 33 HbSS patients and 35 HbAA healthy controls [12]. In that study, average arginine concentration was found to be 72.9 μmol/L, 58.5 μmol/L, and 52.3 μmol/L in healthy controls, asymptomatic patients, and in patients suffering from vaso-occlusive crises, respectively. ADMA was found to be significantly higher in asymptomatic patients (0.70 μmol/L) compared to controls (0.39 μmol/L), without further significant modification during the crisis (0.66 μmol/L). SDMA also increased in asymptomatic patients (0.55 μmol/L) compared to healthy controls (0.42 μmol/L), without further significant modification during the crisis (0.51 μmol/L). Average values obtained in the present study during the crisis (arginine 54.4 μmol/L, ADMA 0.65 μmol/L, SDMA 0.85 μmol/L) and in the steady state (arginine 64.1 μmol/L, ADMA 0.74 μmol/L, SDMA 0.88 μmol/L) are comparable with that previous study [12]. However, our multivariate statistical model reveals that DMA concentration was actually reduced during the crisis, suggesting that DMA levels were no longer homeostatic.

In the steady state of sickle cell disease, it was shown that ornithine and citrulline plasmatic concentration was not modified unlike proline concentration although proline can be synthetized by arginine catabolism [9]. The significant decrease of these amino acids in the metabolomics signature of the crisis suggests that both arginase and NO synthases activities (which sustain ornithine and proline production, and NO and citrulline, respectively) are inhibited. To our knowledge, potential alterations in the plasmatic concentration of other amino acids in the steady state have not been documented. As such, a comprehensive characterization of amino acid concentration associated with crises and how it compares to steady state and healthy individuals is still required.

### 4.2. Red Blood Cell Metabolic Signature

In erythrocytes, compared to the metabolomic profile previously reported in steady state [5], we found a decrease in both aspartate and glutamine concentration during the crisis, while the former is higher and the latter is lower in the steady state compared to healthy individuals. There were no further changes in the concentration of other amino acids (glycine, serine, alanine, ornithine, aspartate, lysine) during the crisis, whereas they have been reported to be altered in the steady state. It should be noted that spermine and spermidine, two polyamines produced from ornithine that are known to stabilize the red blood cell plasma membrane [13,14], are more abundant in red blood cells in the steady state; here, we found no further change during the crisis.

The red blood cell signature associated with the vaso-occlusive crisis was also characterized by an important reconfiguration of phospholipid composition, with an increase in phosphatidylcholines, lysophosphatidylcholines, and four sphingomyelins. This profile differs considerably from that reported in the steady state because a decrease in phosphatidylcholines has been described previously despite an increase in lysophosphatidylcholines [15]. These changes suggest that a remodeling of the red blood cell plasma membrane takes place during the crisis, probably associated with acute sickling. Interestingly, this lipid remodeling impacted on the plasma composition because, in contrast to erythrocytes, there was a significant decrease in many phosphatidylcholines and lysophosphatidylcholines.

### 4.3. Potential New Sickle Metabolites

The present study also suggests that several key metabolic pathways that had not been previously associated with sickle cell disease were altered. In fact, we found an increase not only in plasmatic hexose concentration, but also in phenylalanine and α-aminoadipate. Furthermore, plasmatic DOPA, serotonin, and hydroxyproline decreased, the latter two also being diminished in the erythrocytes. The increase in hexose, along with the alteration of several acyl-carnitines, observed in both the plasma and red blood cells, points to a decrease in the metabolic flux associated with oxidative catabolism, probably due to hypoxia. The increase in α-aminoadipate in the plasma of patients is also consistent with a deregulation of glucose homeostasis and carbohydrate metabolism during the vaso-occlusive crisis. Indeed, α-aminoadipate is an intermediate of lysine catabolism that has been reported to be a predictive biomarker of diabetes [16]. Our findings are thus consistent with the relationship that has been found between sickle cell disease and type 2 diabetes [17].

Here, concurrent alterations in other pathways include (i) a decrease in collagen turnover during the acute episode as suggested by the general decrease in hydroxyproline, which has also been found to be increased in the urine of sickle cell patients in the steady state [18]; (ii) an alteration of the phenylalanine metabolism; (iii) an increase in lipid peroxidation caused by the drop in serotonin. In fact, serotonin has been shown to be able to mitigate lipid peroxidation [19,20,21]. It should also be noted that serotonin liberation by blood platelets exerts a vasoconstricting effect [22], and the change in concentration found here might be involved in the pathophysiology of the vaso-occlusive crisis; and (iv) lastly, an alteration of DOPA metabolism.

### 4.4. Metabolites Potentially Associated with Pain

Acute vascular occlusion is associated with intense pain, which is typical of the crisis. In the literature, many metabolites have been found to be involved in pain pathophysiology. Here, we cross-referenced the keyword “pain” with each of the metabolites identified in our plasma signature using an online search engine. Interestingly, at least 15 of the metabolites found to be significantly changed during the crisis are often related to pain: ADMA, SDMA, total DMA, ornithine, glutamate, glutamine, aspartate, glycine, phenylalanine, α-aminoadipate, serotonin, DOPA, phosphatidylcholines, lysophosphatidylcholines, and sphingomyelins. For instance, serotonin is an important mediator of the descending anti-nociceptive system in the brain stem. In a mouse model of sickle cell disease, it has been proposed that the decline in serotonin was involved in hyperalgesia, thus suggesting serotonin can be used for pain treatment in this disease [23]. Glutamine (found here to decrease in the plasma) is related to both pain and NO metabolism and has been shown to mitigate pain in a phase III clinical trial in sickle cell disease [24]. Arginine metabolism also plays a key role in pain in sickle cell disease [10]. In effect, arginine supplementation has been shown to reduce pain via increased blood NO levels in a double-blind clinical trial [25]. Interestingly, citrulline given orally during steady states in a pilot phase II clinical trial also showed dramatic improvements in symptoms of well-being, while increasing plasma arginine levels [26].

Future clinical trials such as these could take advantage of metabolomics because our study shows that key metabolites could be analyzed and thus implemented for patient monitoring. This growing interest in metabolomics has been highlighted recently by two studies performed before and after red blood cell exchange transfusion in sickle patients, showing a reduction of hypoxia and an improvement of glycolysis together with modifications in amino acids and acyl-carnitine concentration [27,28]. Obviously, it is not possible to exclude the possibility that analgesic treatments taken by patients during the crisis may also contribute this metabolic signature.

## 5. Conclusions

The present study is, to our knowledge, the first report of a metabolomics signature of the vaso-occlusive crisis in sickle cell patients. When compared to the steady state, some metabolite changes were amplified, whereas others were suppressed or remained unaltered (summarized in Figure 6). In the plasma, a striking feature of the metabolic signature was the alteration of NO metabolism and thus potential relationships with pain. In red blood cells, changes in phospholipids were considerable, showing strong membrane remodeling during the crisis. Our study further highlights the involvement of metabolites that have not been found to be involved in sickle cell disease before. In particular, this was the case for hexoses, acyl-carnitines, and α-aminoadipate, which contribute to explaining how the sickle cell disease relates to energetic metabolism.

## Figures and Tables

**Figure 1 jcm-09-01092-f001:**
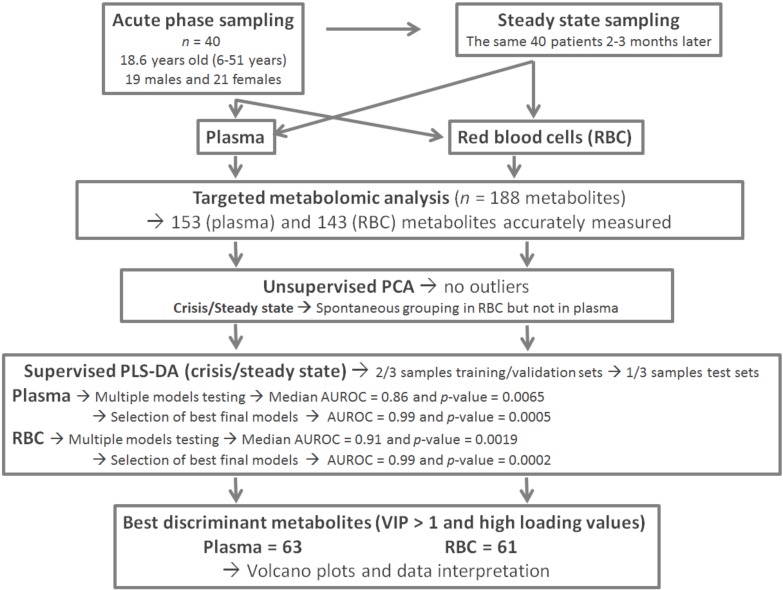
Global flow chart and study design.

**Figure 2 jcm-09-01092-f002:**
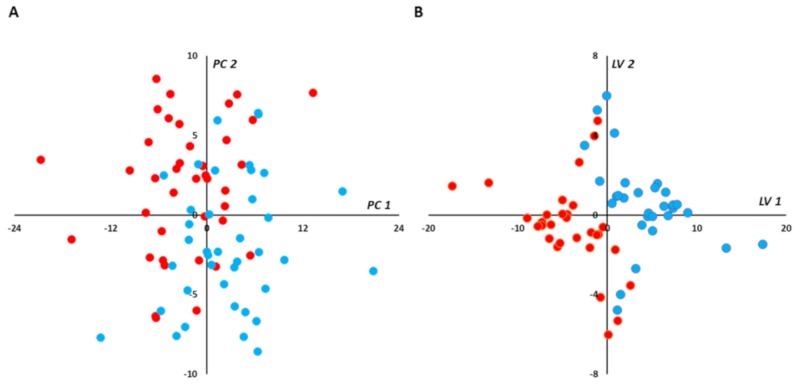
(plasma): Multivariate unsupervised (**A**) and supervised (**B**) scatter plots constructed with plasma samples. (**A**) Principal component analysis (PCA) scatter plot built with all samples (39 in-crisis/39 out-of-crisis). No spontaneous grouping or outliers were detected in the first principal plan. (**B**) Partial least squares-discriminant analysis (PLS-DA) scatter plot built using median coordinates of best models and the 60 training set samples (30 in-crisis/30 out-of-crisis). A good separation was observed in the plan determined by two latent variables with only two samples misallocated. *Legend*: PC1,2: principal component 1 and 2, respectively. LV1,2: latent variable 1 and 2, respectively. Red circles correspond to samples drawn during vaso-occlusive crisis and blue circles to samples drawn after the crisis. Principal components and latent variables have no dimension (arbitrary units).

**Figure 3 jcm-09-01092-f003:**
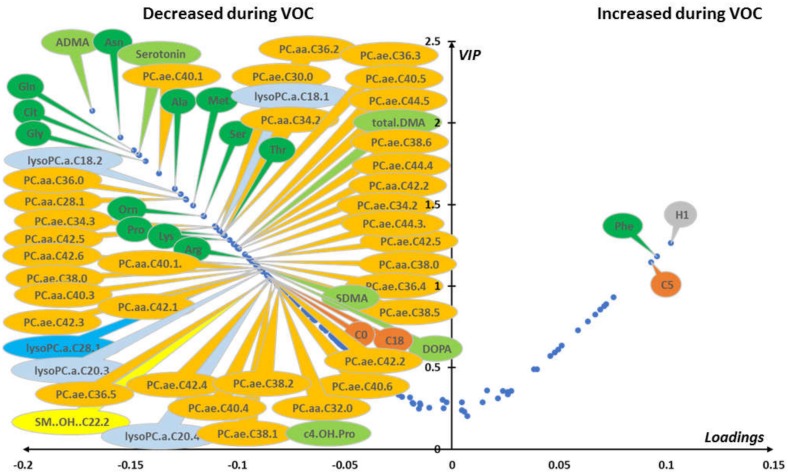
(plasma): Loading vs. variable importance for the projection (VIP; volcano plot) for PLS-DA models constructed from plasma samples. Median loading and VIP were calculated for each metabolite using 4277 best models (BMs) (i.e., AUROC in validation sets (AUROCva) (≥ 0.95). Only the most important metabolites, that is, those with VIP ≥ 1, have been labeled. Positive/negative loadings indicate increased/decreased metabolite concentrations during in-crisis compared to out-of-crisis phases. The main features of the metabolomic signature included relatively increased level of hexoses (H1), phenylalanine (Phe), and valeryl-carnitine (C5) during the crisis. On the other hand, the following metabolites were relatively decreased during the crisis: free carnitine (C0) and octadecanoyl-carnitine (C18); 12 amino acids including asparagine (Asn), glutamine (Gln), glycine (Gly), citrulline (Cit), alanine (Ala), proline (Pro), threonine (Thr), arginine (Arg), lysine (Lys), serine (Ser), methionine (Met), and ornithine (Orn); 6 biogenic amines including asymmetric dimethylarginine, symmetric dimethylarginine, and their sum (ADMA, SDMA, and total DMA, respectively), serotonin (Serotonin), 3,4-dihydroxyphenylalanine (DOPA), and cis-4-hydroxyproline (c4-OH-Pro); 5 lysophosphatidylcholines; 12 diacyl phosphatidylcholines (PC aa) and 22 alkyl-acyl phosphatidylcholines (PC ae); and 1 hydroxy sphingomyelin with an acyl chain of 22 carbons and 2 unsaturated bonds (SM.OH.22.2). Color code for metabolite bubbles: hexose: gray; amino acids: green; biogenic amines: light green; (acyl) carnitines: brown; lysophosphatidylcholines with less than 22 carbons on their acyl group: light blue; lysophosphatidylcholine with more than 22 carbons on its acyl group: dark blue; phosphatidylcholine: orange; hydroxy-sphingomyelin: yellow. For phosphatidylcholines, the sum of the length of the two acyl or acyl-alkyl groups is noted after the “C” and is followed by the number of double bonds. The same notation is used for representing the length and the number of double bonds in the acyl chain of lysophosphatidylcholines. VOC: vaso-occlusive crisis.

**Figure 4 jcm-09-01092-f004:**
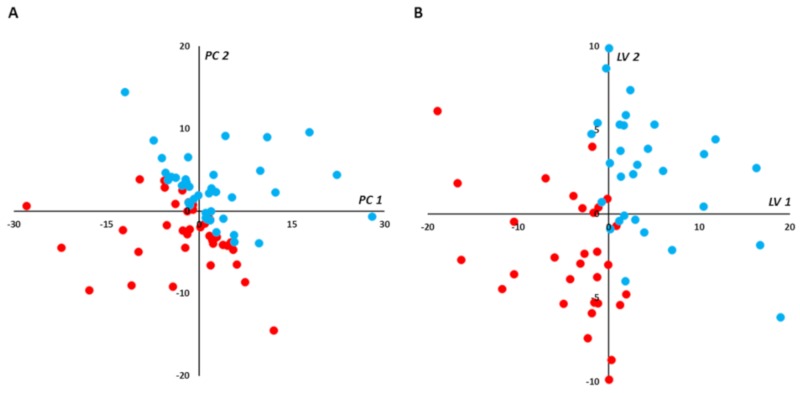
(red blood cells): Multivariate unsupervised (**A**) and supervised (**B**) scatter plots constructed with red cell samples. (**A**) Principal component analysis (PCA) scatter plot built with all samples (*n* = 80; in-crisis (IC)/out-of-crisis (OC) = 40/40). Samples from IC patients (red circles) are roughly separated from samples taken from OC patients (blue circles) in the first principal plan of the PCA. No outlier was detected in this plan. (**B**) PLS-DA scatter plot built using median coordinates of best models and 60 samples (IC/OC = 30/30). A good separation was observed in the plan determined by two latent variables with only two samples misallocated. *Legend*: PC1,2: principal component 1 and 2, respectively. LV1,2: latent variable 1 and 2, respectively. Principal components and latent variables have no dimension (arbitrary units).

**Figure 5 jcm-09-01092-f005:**
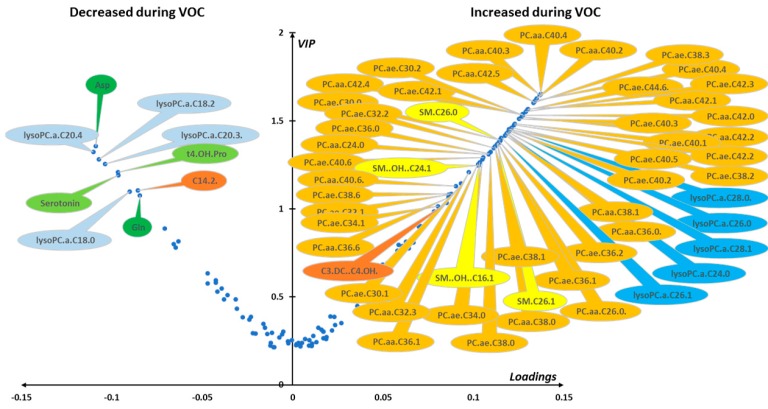
(red blood cells). Loading vs. VIP (volcano plot) for PLS-DA models constructed from red cell samples. Median loading and VIP values were calculated for each metabolite using 7955 best models (BMs) (i.e., AUROCva ≥ 0.95). Only the most important metabolites, that is, those with VIP ≥ 1, have been labeled. Positive/negative loadings indicate increased/decreased metabolite concentrations between the in-crisis and out-of-crisis situation. The main features of the metabolomic signature are a relatively increased level during the crisis of 42 phosphatidylcholines (PC) with 17 diacyl phosphatidylcholines (PC aa) and 25 alkyl-acyl phosphatidylcholines (PC ae), 4 sphingomyelins (SM), malonyl-carnitine (C3-DC), and 5 lysophosphatidylcholines species (LPCs) with acyl chain length longer than 22 carbons. On the other hand, four LPCs with acyl chain length of less than 22 carbons were relatively decreased during the VOC along with aspartate (Asp), glutamine (Gln), trans-4-hydroxyproline (t4-OH-Pro), serotonin, and tetradecadienyl-carnitine (C14:2). Color code for metabolite bubbles: amino acids: green; biogenic amines: light green; acyl-carnitines: brown; lysophosphatidylcholines with less than 22 carbons on their acyl group: light blue; lysophosphatidylcholine with more than 22 carbons on the acyl group: dark blue; phosphatidylcholine: orange; sphingomyelins: yellow. For phosphatidylcholines, the sum of the length of the two acyl or acyl-alkyl groups is noted after the C and is followed by the number of double bonds. The same notation is used for representing the length and the number of double bonds in the acyl chain of lysophosphatidylcholines and (hydroxy) sphingomyelins. VOC: vaso-occlusive crisis.

**Figure 6 jcm-09-01092-f006:**
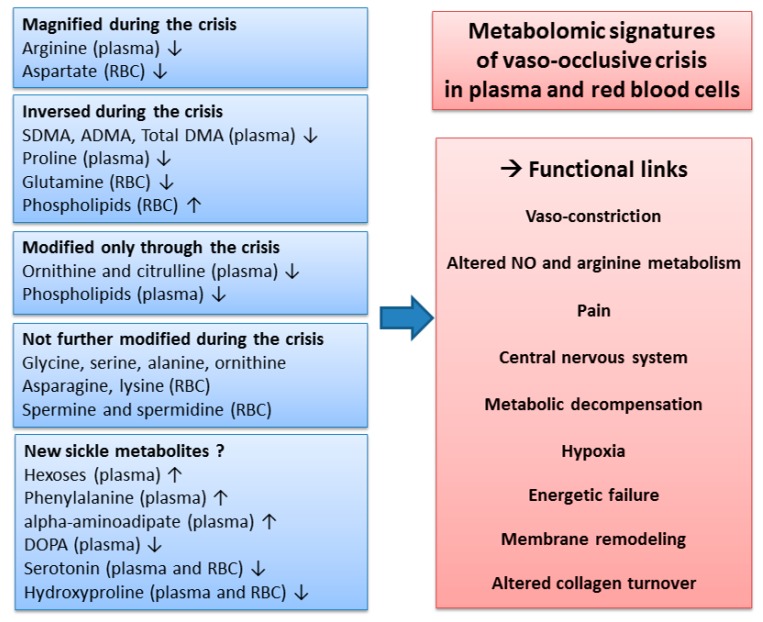
Overview of the metabolomic signature, with potential functional links, of the vaso-occlusive crisis, in plasma and red blood cells. The changes in metabolite concentrations during the crisis observed in our study were compared to the changes reported in the literature during the steady state. ↑: increased concentrations; ↓: decreased concentrations.

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
