# Peer review of "Sickle Cell Disease: Metabolomic Profiles of Vaso-Occlusive Crisis in Plasma and Erythrocytes"

_jcm, 2020, doi:10.3390/jcm9041092_

Round 1

Reviewer 1 Report

This is an interesting exploration of potential markers of crisis in sickle cell disease, which has not previously been well characterized or compared to baseline. The rationale is sound, and the methodology and statistical analysis appropriate. The elucidation of a metabolic signature of crisis in interesting. The correlation between this signature and current therapies (line 389) helps to provide some hope that this could ultimately impact therapy.

I do have some concern about the conclusions being drawn regarding phenylalanine (line 364) and DOPA (line 373). The description of the findings is clear, but do not seem significantly robust to support the conclusions presented. This is a contrast to serotonin, where the correlation is clearly drawn. I would suggest that the results should be presented without the suggested conclusion.

I also find Figure 6 confusing – the headings on the left column all refer to the ‘crisis/steady’ state, which I have taken to mean as crisis state compared to steady state reference. Would be clearer to simply state that in the title or legend of the figure. As well, the arrows drawn between the two columns do not actually indicate correlation between the left and right boxes, but seems to imply that there are linear connections. Would be best to have a single arrow, or none, and leave the 2nd column titled ‘functional links’.

Reviewer 2 Report

This very interesting study advances our understanding of metabolic pathways involved in or impacted by vaso-occlusion in sickle cell disease. I had a number of minor comments to improve this manuscript.

1.       Both in the introduction (line 41), and in the discussion (lines 313, 336, 366) the word “seizure” is used. I assume this is a mis-translation; the correct word in English would be “crisis” or “painful episode”.

2.       The sentence (line 45-47), “The management of the vaso-occlusive crisis consists of fighting the hemoglobin S polymerization, mainly by blood transfusion, hyperhydratation, reoxygenation and appropriate analgesia.”, is not completely accurate. These supportive therapies listed could have some impact on reversing vaso-occlusion, but are not likely to reverse sickle hemoglobin polymerization. Of note, “hyperhydratation” is properly spelled as “hyperhydration”.

3.       To assist others in replicating this study it would be helpful to better describe what was considered to be “steady state” (line 91). Absence of pain, fever, recent blood transfusion? Usual state of health which might include chronic pain?

4.       The authors appropriately describe their results in the context of therapeutic use of glutamine and of arginine, perhaps they could also include a discussion of ongoing studies of citrulline supplementation (Waugh, et al (2001). Oral citrulline as arginine precursor may be beneficial in sickle cell disease: early phase two results. Journal of the National Medical Association, 93(10), 363)?
